# Spatial and multilevel analysis of sanitation service access and related factors among households in Ethiopia: Using 2019 Ethiopian national dataset

Addisalem Workie Demsash[1]*, Masresha Derese Tegegne[2], Sisay Maru Wubante[2], Agmasie Damtew Walle[1], Dereje Oljira Donacho[1], Andualem Fentahun Senishaw[3], Milkias Dugassa Emanu[4], Mequannent Sharew Melaku[2]

1 Department of Health Informatics, College of Health Science, Mettu University, Mettu, Ethiopia,
2 Department of Health Informatics, Institute of Public Health, University of Gondar, Gondar, Ethiopia,
3 Department of Health Informatics, College of Health Science, Debre Markos University, Debre Markos, Ethiopia, 4 Department Of Nursing, College of Health Science, Mettu University, Mettu, Ethiopia

* addisalemworkie599@gmail.com

**Data Availability Statement:** The dataset used for analysis is available on the Measure DHS program

## Abstract

### Background

Billions of people have faced the problem of accessing appropriate sanitation services. This study aimed to explore the spatial distribution of households' access to sanitation services and identify associated factors in Ethiopia.

### Methods

The 2019 Ethiopian Mini Demographic and Health Survey data was used with a total of 6261 weighted samples. A cross-sectional study design with a two-stage cluster sampling technique was used. Global Moran's I statistic measure, Getis-Ord Gi*, and the ordinary Kriging Gaussian interpolation were used for spatial autocorrelation, hot spot analysis, and interpolation of unsampled areas, respectively. A purely spatial Bernoulli-based model was employed to determine the geographical locations of the most likely clusters. A multilevel logistic regression model was used, and predictors with a P value of less than 0.05 with a 95% CI were considered significant factors.

### Results

Overall, 19.7% of households had access to improved sanitation services in Ethiopia. Poor sanitation service access was significantly clustered, with hotspots of poor access identified in the South Nations Nationality and People's Region (SNNPR), Oromia, Amhara, and Benishangul Gumuz regions. A total of 275 significant clusters were identified. Households in the circled area were more vulnerable to poor sanitation service access. Rural households, on-premises water access, media exposure, and rich wealth status were statistically significant factors for access to sanitation services.

(http://dhsprogram.com) website publicly. All the data generated and analyzed during the study are included in the form of maps, tables, and texts in this article.

**Funding:** The authors received no specific funding for this work.

**Competing interests:** The authors have declared that no competing interests exist.

## Conclusions

Access to sanitation services among households in Ethiopia is insufficient. The majority of the households had no access to sanitation services. Stakeholders are recommended to raise household members' awareness of sanitation services, give priority to the hotspot areas, and encourage poor households to have access to toilet facilities. Household members recommended using the available sanitation service and keeping the sanitation service clean. Households are recommended to construct clean shared sanitation facilities.

## Introduction

Sanitation at the household level is critical for good health, disease prevention, and the development of the country [1]. Sanitation service access is a basic human right and an indicator of social and psychological well-being [2]. Worldwide, over 2.5 billion people cannot access improved sanitation services [3]. About 3.6 billion people lack access to safely managed sanitation services, and 673 million people practice open-field defecation [4].

In low-income countries, 62% of urban residents live in awful sanitation conditions [5]. Almost half the population doesn't have access to sanitary facilities [3,6], and only 24% of the rural population is using improved sanitation facilities [7]. Although progress has been made toward achieving basic sanitation coverage, only 10% of countries are on track [8].

According to the 2011 Ethiopia Demographic and Health Survey (EDHS), 82% of households use unimproved toilet facilities [7]. Privately improved sanitation services are used by only 8% of households [9]. In addition to having access to unimproved sanitation services, 9% of households used shared toilet facilities, and 32% of households did not have access to an improved sanitation service [10]. Sanitation service access in Ethiopia isn't adequate. For instance, 87% of the households in Jimma town have used unsafe sanitation services [11]. 45% of available latrine facilities are poor in hygiene, and the pipelines are poor in function in Negele town [12]. In Gondar town, 67% of the households have an unimproved sanitation status, and 51.7% of households have poor hygiene practices [13].

Poor sanitation services access and utilization, a contaminated environment, and water may make it useful to localize specific areas with greater sanitation deprivation, where sanitation intervention factors are prioritized for reducing diarrhea-related morbidity and mortality and serious diseases like cholera and typhoid [14]. Inappropriately designed and inaccessible sanitation facilities are problems for physically disabled people [15]. Different individual and community-level factors may represent associated factors for sanitation service access. Family size, the sex of the household head, occupation, income, wealth status, toilet age [11], and educational and marital status [12] may be associated factors for sanitation service access.

Due to the demonstrated impact of a lack of basic sanitation access on public health, especially in underdeveloped countries, this paper stands out for highlighting the problems and identifying areas where these deficits are increasing and are major factors that lead to health inequities. Recognizing these areas would improve the cost-benefit access to interventions, increasing the population reached by interventions in resourceslimited setting. Exploring the spatial distribution of households' access to sanitation services in Ethiopia could help identify and prioritize sanitation interventions in specific areas with the greatest need. So, this study aimed to describe sanitation service access spatially, identify hotspot areas, and identify factors associated with households' sanitation service access.

## Methods

### Study design and setting

A nationally representative cross-sectional study design was used. This study was done across nine regions of Ethiopia, including Addis Ababa and Dire Dawa city administrations.

### Data source

The 2019 EMDHS dataset was used from the DHS website (http://www.measuredhs.com). Shapefiles were downloaded from the African Open Data website (https://africaopendata.org/dataset).

### Study period

The 2019 EMDHS data was collected from March 21, 2019, to June 28, 2019, by the Ethiopian Public Health Institute in collaboration with the Central Statistical Agency to generate data for measuring the progress of the health sector goals set under the Growth and Transformation Plan.

### Sampling producers

A two-stage stratified cluster sampling was used. Each region was stratified into urban and rural areas, yielding 21 sampling strata. At the first stage of selection, 305 enumeration areas (EAs) were chosen at random with a probability proportional to each EA. To ensure the survey is comparable across regions, 25 EAs were selected from eight regions (25*8 = 200 EAs), and 35 EAs were selected from each of the three large regions such as Amhara, Oromia, and SNNPR (35*3 = 105 EAs) with equal proportional sample allocation. In the second stage of selection, a fixed number of 30 households per cluster were selected with equal probability through systematic selection.

### Populations

All women aged 15–49 years in the households were source populations, whereas all women aged 15–49 years who were either permanent residents or visitors in the sampled households who slept in the households the night before the survey were the study population. Zero coordinates and clusters that had undefined proportional access to sanitation services were excluded since the hotspot, SatScan, and interpolation analysis were performed based on the proportion of cases and controls. The details of the methodology are available from the 2019 EMDHS report [16].

### Study variables and their measurements

The study's outcome variable was access to sanitation services. The households had improved access to sanitation services if they had access to at least one of the following: flush or pour-flush to a piped sewer system, septic tank, pit latrines, ventilated-improved pit latrines, pit latrines with a slab, or composting toilets. The household had non-improved (poor) sanitation service access, If the household had any of the following toilet types: flush somewhere else; pit latrines without slabs; open pits and buckets; hanging toilets; or open field defecation, including no facility, bush, or field, it was considered to have poor sanitation service access [16,17].

The independent variables, such as sex and age of household heads, media exposure, wealth status, shared facilities, and size of the family unit of the households, were used as individual-

level independent variables. Community-level predictor variables in this study included the place of residence, region, placement of toilet facilities, and time to reach the water source.

Shared toilet facilities were assessed whether one household share the available toilet facilities with another household. So, if the households had shared the toilet facilities with another households it was labelled as yes = 1, else no = 0 [18,19]. The wealth status was generated from the wealth index for the households. In the dataset, the wealth index has five quintiles, such as the lowest quintile (poorest), the second quintile (poorer), the third quintile (middle), the four quintiles (rich), and the fifth quintile (richest). For ease of analysis, the first and second wealth index categories, such as "poorest" and "poorer" were labeled as 'poor = 1', and the middle wealth index category was labeled as 'middle = 2', whereas the fourth and fifth wealth index categories, such as "rich" and "richest" were labeled as 'rich = 3' [20].

Media exposure was defined as access to the media that might help households or household members access information or messages related to sanitation or hygiene. Therefore, if households had had either radio, television, or both (radio and television), then households had had media exposure. Otherwise, households had no media exposure [21,22]. The protected water source was defined as access to a protected water source if the households have access to piped water, whether it be piped into the dwelling/yard/neighborhood, standpipe water, public well water, borehole water, protected well or spring water, rain, or bottled water [23,24].

## Data management and processing

Data cleaning, labeling, and processing were done using STATA version 15 software and Microsoft Office Excel. For accurate parameter estimations and representativeness, sample weighting was performed. The descriptive analysis results were presented in a table and text narration to describe the study subject and poor sanitation service access among households.

## Spatial data analysis

**Global spatial autocorrelation and hot spot analysis.** Arc Map version 10.7 software was used for spatial autocorrelation and the detection of hot spot areas with poor sanitation service access. Global Moran's I statistic measure was used to assess whether poor access to sanitation services was dispersed, clustered, or randomly distributed in Ethiopia [25]. Moran's I value is close to minus one (-1), close to plus one (+1), or zero (0), indicating a dispersed, clustered, and random distribution of households' poor sanitation service access, respectively [23,24]. Poor sanitation service access among households is determined by the z scores and significant p-values of the hot spot analysis (Getis-Ord Gi*).

**Spatial interpolation.** Households' poor sanitation service access in unsampled areas was predicted by using the spatial interpolation technique. To predict households' poor sanitation service access in the unsampled areas, the existing evidence of poor sanitation service access was used as input. To minimize prediction uncertainty, an ordinary Kriging Gaussian interpolation technique was employed. Based on the input data at each location, a semi-variogram model was constructed and used to define the weight that further determines the prediction of new values in unsampled areas [26,27]. As a result, based on a simulated semi-variogram, interpolation model (map) was generated.

**Spatial scan statistics.** A SatScan version 9.5 software was used for local cluster detection analysis [28]. Purely spatial Bernoulli-based model scan statistics were employed to determine the geographical locations of statistically local significant clusters with high rates of household sanitation service access [29]. Those households that had access to improved sanitation service were taken as cases, and those households that had no access to improved sanitation service

were taken as controls to fit the Bernoulli model. The default maximum spatial cluster size <50% of the population was considered to allow small and large clusters to be detected, and to ignore clusters that contained more than the maximum limit due to the circular shape of the window. A log-likelihood ratio test statistic was used to determine if the number of observed cases within the potential cluster was significantly higher than expected or not. The circle with the maximum likelihood ratio test statistic was defined as the most likely cluster, then compared with the overall distribution of maximum values. The primary and secondary clusters were identified, assigned p values, and ranked based on their likelihood ratio test on the basis of the 999 Monte Carlo replications [30].

## Multilevel logistic regression analysis

Multilevel mixed-effect logistic regression analysis was conducted. Respondents were nested within households, and households were nested within clusters. Therefore, respondents from the same cluster had more similarity than those respondents who were from another cluster inters of the outcome of interest. So, data dependency might be existed. To alleviate correlations between the clusters, we assumed four models: **model 1** (a null model that assesses the dependency of poor sanitation service access across the cluster), **model 2** (contains individual-level variables), **model 3** (community-level variables), and **model 4** (aggregate model of **models 2** and **3**). For each model, the Intraclass Correlation Coefficient (ICC) was calculated to check whether the data is eligible for multilevel mixed-effect logistic regression or not.

Consequently, 70% of ICC's values (**Fig 1**) confirmed that there was spatial significant correlation in the households' access to sanitation service access within clusters, and so multilevel mixed-effect logistic regression analysis was fitted to assess both individual and community level variables in the access of sanitation services. Multicollinearity was assessed using the Variance Inflation Factor. Hence, the value of Variance Inflation Factor was 2.5 which indicated there is no any multicollinearity between predictors. In multilevel mixed-effect logistic regression analysis, a p-value less than 0.05 with a 95% CI was used to identify associated factors of households' poor sanitation service access.

## Ethics approval

Ethical approval and consent from study participants were not necessary for this study because it was based on publicly available data from the Measure DHS program website

| Measure of variations | Model 1 | Model 2 | Model 3 | Model 4 |
|---|---|---|---|---|
| Variance | 0.93 | 0.40 | 0.22 | 0.177 |
| ICC | 0.70 (.64, .75) | 0.46(.39, .53) | 0.33(.27, .39) | 0.26(.21, .33) |
| LLR | -2392.98 | -2261.50 | -2245.91 | -2166.43 |
| MOR | 7.56 (5.94, 9.63) | 2.76 (2.08, 3.65) | 1.58(1.2, 2.08) | 1.18 (.88, 1.58) |
| AIC | 4789.96 | 4561.00 | 4517.83 | 4392.86 |

**Fig 1. Model comparisons.**

(https://dhsprogram.com), permission was obtained to access the EMDHS data for statistical analysis and reporting.

## Results

### Sociodemographic characteristics

Four out of ten (38.10%) and nearly one-fifth (23.30%) of households were from the Oromia and SNNPR regions, respectively. Six out of ten (61.80%) households were rural residents. The majority (78.70%) of household heads were males, and approximately four to ten (38.00%) of household heads were under the age of 15–35 years. More than half (56.40%) of households were rich. One-half (50.70%) and seven to ten (68.20%) of households had at most four family members and had not shared toilet facilities with other households, respectively. The majority (68.50%) of households had improved access to drinking water sources. A majority of households do not have access to radio (67.60%) and television (77.50%) (**Table 1**).

**Table 1. Sociodemographic characteristics.**

| Variable | | Frequency (n) | Percent (%) |
|---|---|---|---|
| Region | Tigray | 302 | 4.80 |
| | Afar | 31 | .50 |
| | Amhara | 1418 | 22.70 |
| | Oromia | 2387 | 38.10 |
| | Somali | 149 | 2.40 |
| | Benishangul | 79 | 1.30 |
| | SNNPR | 1440 | 23.00 |
| | Gambela | 21 | .30 |
| | Harari | 20 | .30 |
| | Addis Ababa | 367 | 5.90 |
| | Dire Dawa | 46 | .70 |
| Age of households' head | 15–35 years | 2378 | 38.00 |
| | 36–50 years | 2021 | 32.30 |
| | >50 years | 1861 | 29.70 |
| Sex of households' head | Male | 4927 | 78.70 |
| | Female | 1334 | 21.30 |
| Wealth status | Poor | 1456 | 23.20 |
| | Middle | 1276 | 20.40 |
| | Rich | 3529 | 56.40 |
| Residency | Urban | 2389 | 38.20 |
| | Rural | 3872 | 61.80 |
| Sharing toilet | No | 4273 | 68.20 |
| | Yes | 1988 | 31.80 |
| Family size | < = Four | 3174 | 50.70 |
| | >Four | 3086 | 49.30 |
| Households have radio | No | 4230 | 67.60 |
| | Yes | 2031 | 32.40 |
| Household has Television | No | 4852 | 77.50 |
| | Yes | 1409 | 22.50 |
| Source of drinking water | Protected | 4287 | 68.50 |
| | Unprotected | 1974 | 31.50 |

## Households' sanitation services access

Overall, 19.70% (95% CI: 18.72%–20.69%) of households had access to improved sanitation services in Ethiopia. From the improved types of toilet facilities, 5.65% and 15.64% of households had used flushing to pit latrine, and pit latrine with slab respectively. the majority of the households (72.24%) had used open-field defection. Nearly one-fifth (26.80%) of households did not have access to sanitation services at all (**Fig 2**).

Spatial distribution of poor sanitation service access among households in Ethiopia

The spatial distribution of poor sanitation service access among households in Ethiopia was nonrandom (Global Moran's I = 0.650999, P-value = 0.000000). The spatial autocorrelation report revealed that the spatial distribution of poor sanitation service access among households was significantly clustered in the regions of Ethiopia within 115.73 kilometers (KMs) of the threshold distance (**Fig** 3).

The red color points indicate the area where households with poor sanitation services access are aggregated or clustered. The hot spots of poor sanitation service access among households were significantly clustered in the Afar, SNNPR, the Western Oromia and Amhara, and the Benishangul Gumuz regions. Cold spots of households' poor sanitation service access were significantly clustered in Dire Dawa and Addis Ababa city administrations, Harari and Tigray regions (Fig 4).

## Spatial SatScan analysis

The colored windows indicate significant clusters of households poor sanitation service access. A total of 233 significant clusters were identified. The 109, 98, and 26 were primary, secondary, and tertiary significant clusters, respectively. In the Gambela, SNNPR, Benishangul Gumuz,

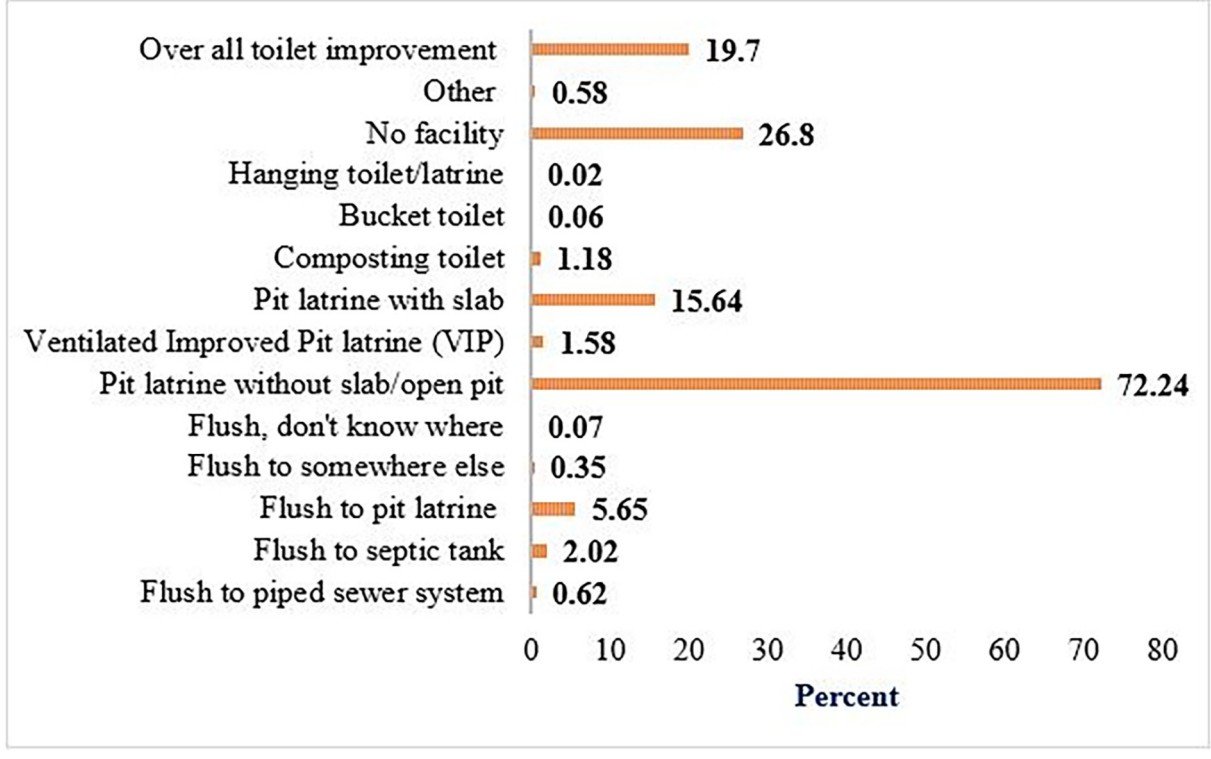

**Fig 2. Sanitation services access of households in Ethiopia.**

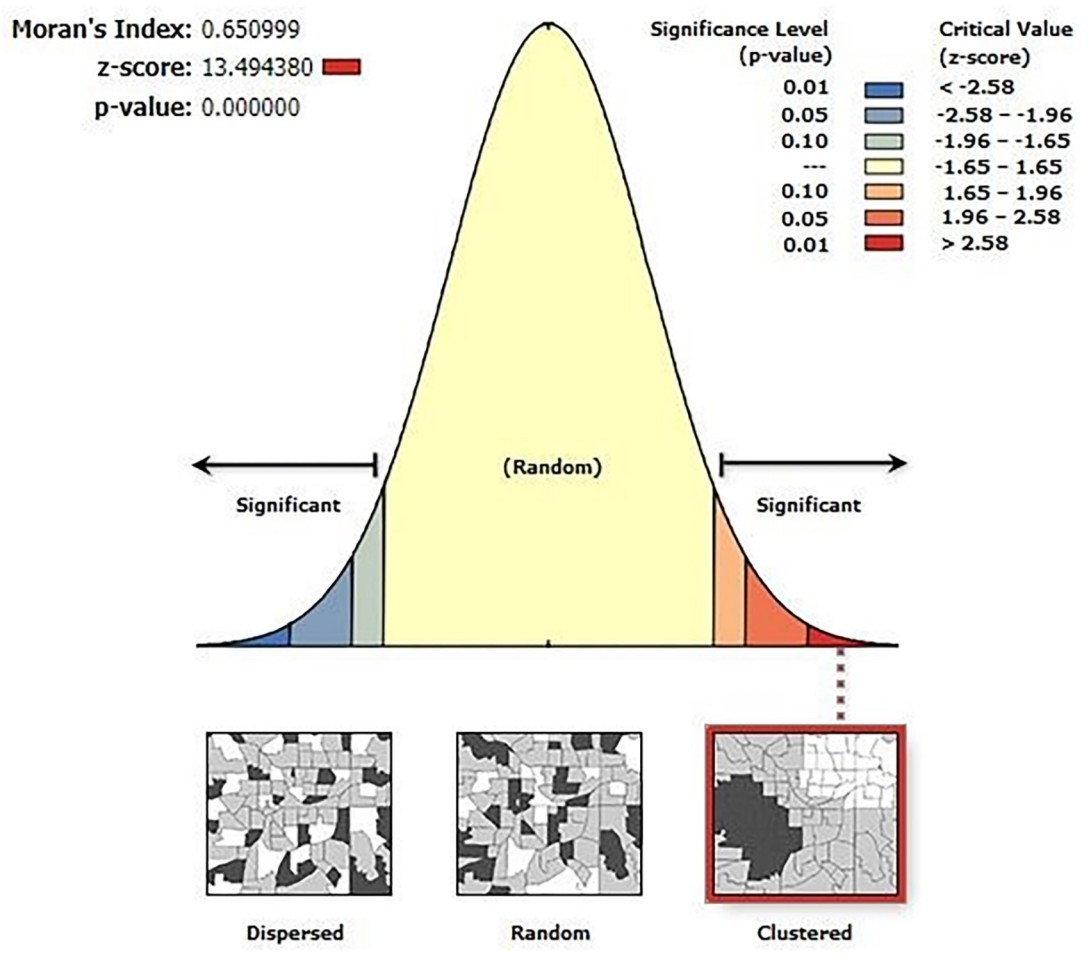

Given the z-score of 13.4943797161, there is a less than 1% likelihood that this clustered pattern could be the result of random chance.

**Fig 3. Spatial autocorrelation report of households' poor sanitation service access in Ethiopia.**

Oromia, and Amhara regions, primary and secondary clusters were located at 8.053039° N, 33.198166° E within a 609.44 KM radius and 6.708096° N, 35.156792° E within a 458.62 KM radius, respectively. The tertiary significant clusters were located at 11.722588° N and 38.322763° E within a 167.41KM radius in the Amhara and southern parts of the Tigray regions. Households in the primary and secondary clusters were 2.21 and 2.16 times more vulnerable to poor sanitation service access than households outside the window respectively (**Table 2, Fig 5**).

## Interpolation of the households' poor sanitation service access

An ordinary Gaussian Kriging interpolation method was employed. The interpolation result indicated that households in the Northern Amhara, Western Gambela, Tigray, Somali, and Harari regions, and households in Addis Ababa and Dire Dawa, would be less vulnerable to poor sanitation service access. However, Afar, SNNPR, Benishangul Gumuz, Oromia, and southern Amhara regions would be more vulnerable to poor sanitation service access (**Fig 6**).

## The red color show the hot spots of poor sanitation services access

**Fig 4. Hotspot analysis of households' poor sanitation service access.**

## Measure of variation

There was a significant correlation and variation among households' access to sanitation services in Ethiopia in each cluster. The intraclass correlation coefficient (ICC) and variance of sanitation service access in **model 1** show that there were 0.70 and 0.93 ICC and variation in

**Table 2. Most significant clusters of spatial SatScan analysis.**

| Types of cluster | Detected cluster | Coordinates/ Radius | Populations | Case | RR | LLR | p-value |
|---|---|---|---|---|---|---|---|
| Primary | 219, 220, 217, 206, 230, 211, 212, 214, 209, 207, 208, 170, 228, 225, 226, 227, 221, 224, 118, 222, 210, 223, 155, 94, 215, 154, 147, 86, 152, 153, 194, 216, 200, 151, 157, 201, 156, 92, 150, 146, 120, 149, 195, 169, 167, 168, 97, 93, 96, 160, 161, 158, 91, 164, 196, 166, 159, 148, 192, 98, 163, 95, 173, 204, 87, 162, 191, 119, 77, 165, 80, 198, 174, 179, 79, 189, 199, 190, 177, 180, 112, 171, 176, 197, 178, 52, 202, 99, 72, 205, 203, 75, 53, 184, 76, 115, 172, 70, 73, 188, 74, 175, 54, 187, 81, 116, 182, 185, 71 | 8.053039N, 33.198166E/609.44KM | 2413 | 2092 | 2.21 | 705.72 | < 0.001 |
| Secondary | 215, 200, 216, 228, 224, 210, 223, 222, 227, 201, 226, 221, 225, 194, 195, 214, 206, 212, 196, 192, 208, 209, 94, 207, 211, 96, 230, 173, 97, 91, 204, 217, 220, 191, 120, 198, 118, 92, 199, 219, 190, 95, 197, 189, 93, 202, 179, 180, 177, 178, 170, 169, 168, 174, 172, 86, 167, 155, 188, 115, 98, 184, 176, 87, 171, 154, 203, 182, 112, 156, 153, 152, 205, 187, 147, 89, 150, 185, 181, 186, 151, 157, 116, 113, 149, 164, 183, 119, 175, 117, 161, 166, 146, 158, 99, 160, 148, 163 | 6.708096N, 35.156792E/458.62KM | 2219 | 1959 | 2.16 | 697.09 | < 0.001 |
| Tertiary | 58, 60, 61, 83,78, 62, 59, 65, 81, 54, 82, 70, 71, 74, 76, 84, 51, 53, 63, 75, 72, 56, 18, 66, 52, 73 | 11.722588N, 38.322763E/167.41KM | 467 | 373 | 1.39 | 48.95 | <0.001 |

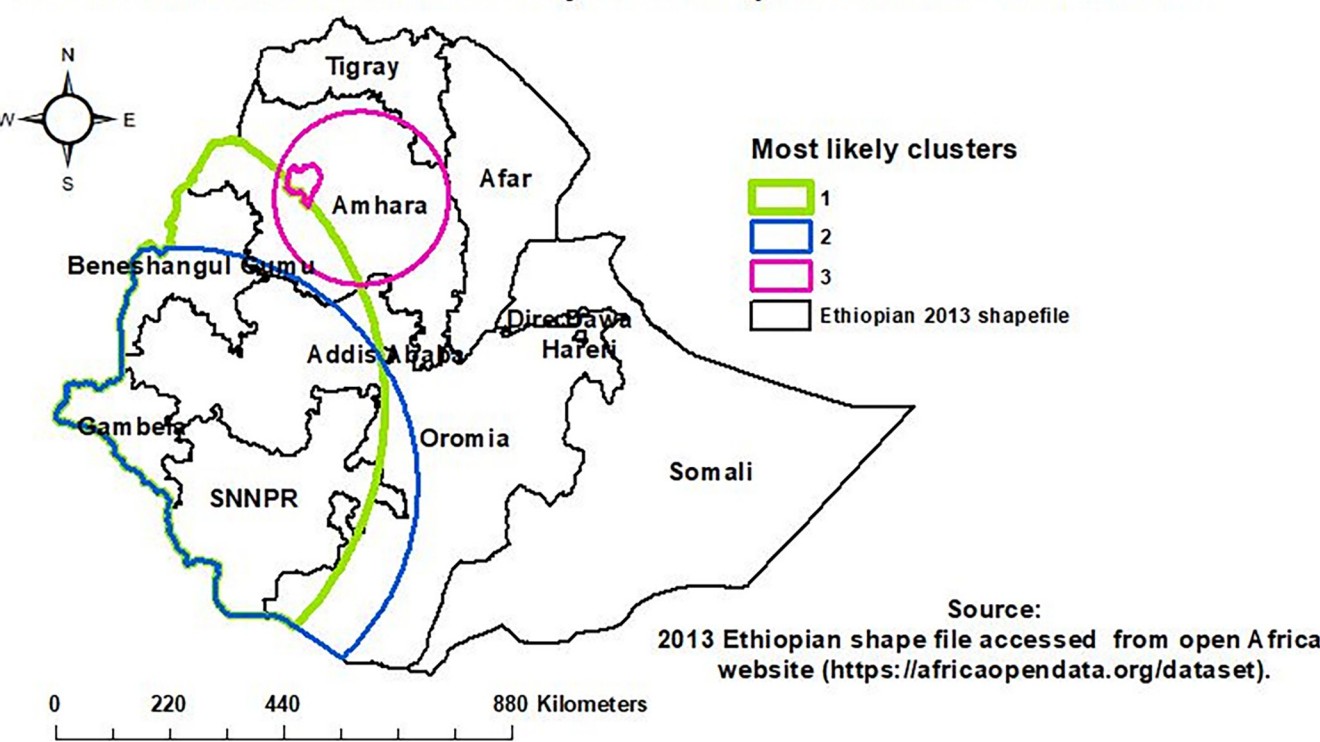

**Fig 5. Sat Scan analysis of households' poor sanitation service access.**

clusters, respectively. This means that there were 70% correlation and 93% reliability variations in households' sanitation services access. Overall model comparisons and the effects of each model were presented in Fig 2.

### Individual and community-level factors influence households' access to sanitation services

In the multilevel mixed-effect logistic regression analysis, sharing a toilet, wealth status, media exposure, time to get a water source, placement of the toilet facility, and rural households were statistically significant factors for appropriate sanitation service access. Households that shared toilet facilities were 27% (AOR: 1.27, 95% CI: 1.04, 1.55) more likely to access appropriate sanitation services than their counterparts. Households exposed to media were 1.9 (AOR: 1.87, 95% CI: 1.46, 2.36) times more likely to access appropriate sanitation services than their counterparts. Rich households were 1.7 (AOR: 1.66, 95% CI: 1.22, 2.26) times more likely to access appropriate sanitation services than poor households. Households that obtain their water on their own premises were 2.6 (AOR: 2.56, 95% CI: 1.91, 3.44) times more likely to access appropriate sanitation services than households that traveled up to 30 minutes to get their water. Households whose toilets were placed in their own yards were 1.2 (AOR: 1.15, 95% CI: 0.92, 1.77) times more likely to access appropriate sanitation services than households whose toilets were placed elsewhere. Rural households had 68% (AOR: .32, 95% CI: .19, .49) less odds of accessing appropriate sanitation services than urban households (Table 3).

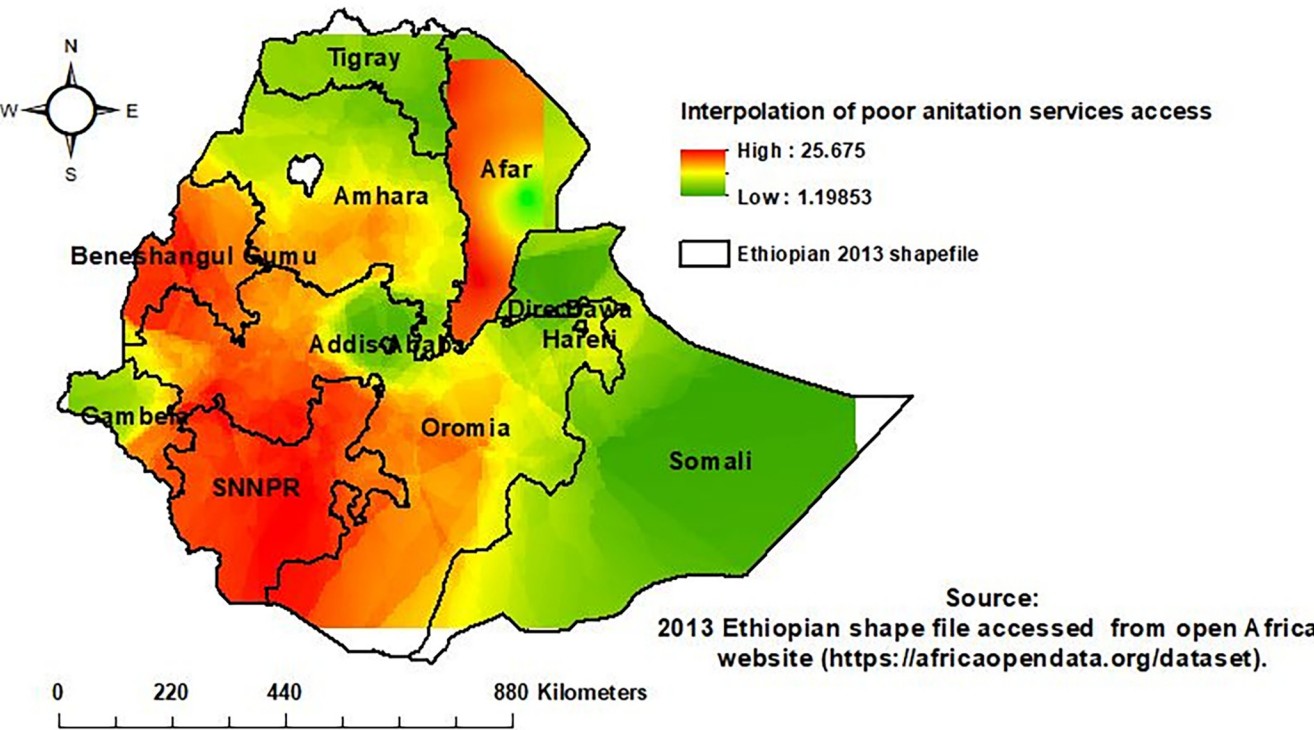

**Fig 6. Interpolation of poor sanitation service access.**

## Discussion

In this study, nearly one-fifth (26.8%) of households didn't have access to sanitation services, and the majority (72.24%) of households exercised open defecation. Overall, 19.7% (95% CI: 18.72%–20.69%) of households had access to improved sanitation services in Ethiopia. This finding is higher than reports in rural Ethiopia (5%) [31,32], the 2016 EDHS report (6%) [33], and Western Africa (8.7%) [34]. However, the finding is lower than studies done in Gondar town (29.2%) [15], Bahir Dar (34%) [35], southern Ethiopia (27.1%) [36], rural Mali (42%) [37], and developing countries (28%) [38]. Generally, there was low sanitation service access among households in Ethiopia. This might be due to governmental and non-governmental organizations' poor attention and attitude towards latrine accessibility and utilization [15], and limited finance to construct latrine facilities [37]. Furthermore, Moreover, households might come to believe that open defecation is preferable to dirty, and poorly ventilated latrines [39], ineffective social mobilization techniques, inadequate support for poor households, a lack of sanitation technology to access advanced and safer forms of sanitation, and a lack of periodic training for an in-depth understanding of the importance of sanitation may all be impediments to low sanitation service access [40,41].

The spatial distribution of households' poor sanitation service access was not random and was significantly clustered in Ethiopia. Poor sanitation service access among households was most prevalent in the Afar, SNNPR, Western Oromia, Northern Amhara, and Benishangul Gumuz regions. Households in Dire Dawa, Addis Ababa, and Tigray regions were less vulnerable to limited access to sanitation services. In a purely Burnell-based model SatScan analysis, a total of 233 significant clusters were identified. Most likely clusters were found in the

**Table 3. Multilevel mixed-effect logistic regression analysis of households' appropriate sanitation survives access in Ethiopia, Using 2019 EMDHS data.**

| Variables | Category | Model 1 | Model 2 | Model 3 | Model 4 |
|---|---|---|---|---|---|
| | | | AOR (95% CI) | AOR (95% CI) | AOR (95% CI) |
| Source water | Protected | | 1.01(.80, 1.26) | - | 1.05 (.84, 1.32) |
| | Unprotected | | 1 | | 1 |
| Household head's age | 36–50 years | | .81(.66, 1.00)[b] | - | .82 (.67, 1.01) |
| | >50 years | | .92 (.73, 1.16) | - | .94 (.75, 1.18) |
| | 1–35 years | | 1 | - | 1 |
| Wealth status | Rich | | 1.93 (1.41, 2.63)[b] | - | 1.66 (1.22, 2.26)[a] |
| | Middle | | 1.0 (.73, 1.38) | - | 1.03 (.75, 1.41) |
| | Poor | | 1 | | 1 |
| Households' head sex | Female | | 0.99(.81, 1.21) | - | .92 (.75, 1.12) |
| | Male | | | | 1 |
| Media exposure | Yes | | 2.35 (1.86, 2.99)[b] | - | 1.87(1.46, 2.36)[a] |
| | No | | 1 | | 1 |
| Family size | >four | | 1.13 (.93, 1.38) | - | 1.15 (.94, 1.40) |
| | < = four | | 1 | | 1 |
| Share toilet | Yes | | 1.43 (1.18, 1.74)[b] | - | 1.27(1.04, 1.55)[a] |
| | No | | 1 | | 1 |
| Region | Afar | | - | 1.42(.5, 3.85) | 1.47 (.57, 3.77) |
| | Amhara | | - | .43(.2, .96) | .6 (.28, 1.26) |
| | Oromia | | - | .15(.05, .26) | .17 (.08, .36) |
| | Somali | | - | 3.1(1.18, 8.14) | 4.11(1.63, 9.83) |
| | Benishangul | | - | .15 (.06, .36) | .23 (.10, .53) |
| | SNNPR | | - | .16 (.1, .36) | .24 (.11, .49) |
| | Gambela | | - | .14 (.05, .36) | .19 ( .07, .46) |
| | Harari | | - | 1.1 (.46, 2.5) | .79 (.36, 1.76) |
| | Addis Ababa | | - | 3.27 (1.3, 8.19)[b] | 2.59 (1.09, 6.18) |
| | Dire Dawa | | - | 4.56 (1.9, 11.0) | 3.59 (1.8, 9.01) |
| | Tigray | | | 1 | 1 |
| Residency | Rural | - | - | .11(.07, .17)[b] | .32 (.19, .49) |
| | Urban | | | 1 | 1 |
| Toilet facility placement | In own yard | | - | 1.35(.99, 1.97)[b] | 1.15 (.92, 1.77) |
| | In own dwelling | | - | 1.29 (.98, 2.45)[b] | 1.19 (.96, 2.02) |
| | Elsewhere | | | 1 | 1 |
| Time to reach to a water source | >30 minutes | | - | 1.38 (1.05, 1.82)[b] | 1.19 (.80, 1.79) |
| | On-premises | | - | 4.05 (2.99, 5.46)[b] | 2.56 (1.91, 3.44)[a] |
| | < = 30 minutes | | | 1 | 1 |

a = Significant at **Model 4**, b = Significant at **Model 2, 3**, 1 = Reference category.

SNNPR, Gambela, Benishangul Gumuz, Oromia, and Amhara regions. The interpolation result indicated that households in the Northern Amhara, Western Gambela, Tigray, Afar, Somali, and Harari regions, as well as in Addis Ababa and Dire Dawa cities, would be less vulnerable to poor sanitation service access. However, SNNPR, Binishangul Gumez, Oroima, and Southern Amhara regions would be more vulnerable to poor sanitation service access The vulnerability of households to poor sanitation service access in Ethiopia might be due to a lack of information [11], low latrine ownership [42], poor latrine conditions, structure, and design [43], and a lack of total sanitation and hygiene intervention led by the community [44].

Individually independent variables, such as sharing toilet facilities, media exposure, and wealth status were significantly associated with households' access to sanitation services. Households that shared toilet facilities were 1.3 times more likely to access appropriate sanitation services than their counterparts. Since improved toilet facilities are impossible for households with high poverty and have limited space for sanitation service construction. So, if properly operated, maintained, and secured, sharing toilet facilities provides significant health benefits as improved toilet facilities for individual households [45]. Furthermore, in densely populated urban areas, sanitation facilities for low-income households may be shared [46]. Furthermore, people may require safe sanitation services while away from home [47]. Sharing sanitation services is common in developing countries, so millions of people might be relied on shared facilities [48].

Households exposed to the media were 1.9 times more likely to access appropriate sanitation services than their counterparts. This finding is supported by a study done in sub-Saharan Africa [49]. This might be because households exposed to media are more likely to access information about aspects of sanitation services, enhance their knowledge towards personal hygiene, build positive attitudes, and value the safe disposal of faeces and stools [49,50].

Rich households were 1.7 times more likely to access sanitation services than poor households. This finding is supported by studies done in Southern Ethiopia [36], the 2016 EDHS analysis report [23], and Zambia [51]. This could be due to the high cost of constructing a latrine, households' inability to pay for labour and construction materials [52], and government policy that does not provide subsidies for residential latrines in comparison to other countries [53]. Therefore, wealthy households have a better chance of constructing and building effective and long-lasting latrines to meet their needs, and they can afford to pay for labor and materials [23].

Households that obtained their water on their own premises were 2.6 times more likely to access appropriate sanitation services than households that and traveled up to 30 minutes to get water sources. This finding is supported by the 2016 EDHS analysis report [23]. This might be due to the length of time it takes to reach (proximity of the house) water sources, which might make a difference in access to improved sanitation services. A nearby septic tank and sewer system, the availability of nearby public toilet facilities, and a physically short distance might be reasons for toilet facility accessibility [23]. **Similarly**, households whose toilets were placed in their own yard were 1.2 times more likely to access appropriate sanitation services than households whose toilets were placed elsewhere. This finding is supported by studies done in northern [42], and northwest Ethiopia [54]. This might be because households need to access and utilize functional toilet facilities within their own privacy, security, and comfort [54,55]. Plus, households might need to build a safe, good-quality, and correctly placed toilet and accept the minimum recommended distance (10 meters) of the toilet from the home [54,56].

Rural households had 68% lower odds of accessing appropriate sanitation services than urban households. This finding is supported by studies done in Indonesia [57], Ghana [58], and Sub-Saharan Africa [49]. This might be tbecause people from rural areas might not have adequate financial resources and so are unable to access improved sanitation services [23], the existence of unfair accessibility of sanitation services [51], and the accessibility of improved water sources in urban areas as compared with rural areas [59].

## Limitations and strengths of the study

There are several limitations to this study. A cross-sectional study analysis may not provide evidence of a causal relationship between the outcome and independent variables. Data on

personal and household practices were based on the mothers' recall, which might have been subject to recall bias. Plus, the analysis did not include all determinants of households' access to sanitation services due to a lack of data detailing these variables in the DHS. Even if efforts were made to reduce the percentage of population detected to 10% to report hierarchical clusters with no geographical overlap, geographical overlap existed among the local clusters of the SatScan analysis local clusters. Despite these limitations, the study's data was collected across the country, making it nationally representative. Furthermore, multilevel analysis was employed, which is more appropriate for cluster data, to solve the data dependency.

## Conclusions

The majority of households in Ethiopia had unimproved access to sanitation services. Wealth status, media exposure, sharing toilet facilities, time to get a water source, placement of toilet facilities, and being in rural areas were statistically significant factors for households to access appropriate sanitation services in Ethiopia. Hence, government officials and health professionals recommended creating awareness among rural household members regarding sanitation services and enhancing the wealth status of poor households. Stakeholders also encourage and formulate standards for households to share the available toilet facilities in appropriate ways. Stakeholders pay priority attention to the hot spots and expose the communities to information about sanitation services through media such as television and radio.

## Acknowledgments

We would like to express our deepest heartfelt thanks to the Measure DHS program for providing the data for this study.

## Author Contributions

**Conceptualization:** Addisalem Workie Demsash.

**Data curation:** Addisalem Workie Demsash.

**Formal analysis:** Addisalem Workie Demsash.

**Methodology:** Mequannent Sharew Melaku.

**Writing – original draft:** Addisalem Workie Demsash.

**Writing – review & editing:** Addisalem Workie Demsash, Masresha Derese Tegegne, Sisay Maru Wubante, Agmasie Damtew Walle, Dereje Oljira Donacho, Andualem Fentahun Senishaw, Milkias Dugassa Emanu.

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
