## [Decision Letter · Decision Letter 0]

4 Jan 2023

PGPH-D-22-01942

Spatial and Multilevel Analysis of Sanitation Service Access and Related Factors in Ethiopia: Using 2019 Ethiopian national data.

Dear  Mrs. Addisalem Workie Demsash,

Thank you for submitting your manuscript to PLOS Global Public Health. After careful consideration, we feel that it has merit but does not fully meet PLOS Global Public Health’s publication criteria as it currently stands. Therefore, we invite you to submit a revised version of the manuscript that addresses the points raised during the review process.

We look forward to receiving your revised manuscript.

Kind regards,

Reginald Quansah, Ph.D.

Academic Editor

Journal Requirements:

1. Please provide separate figure files in .tif or .eps format only and remove any figures embedded in your manuscript file. Please also ensure that all files are under our size limit of 10MB.

2. We do not publish any copyright or trademark symbols that usually accompany proprietary names, eg  ©, ®, ™  (e.g. next to drug or reagent names). Please remove all instances of trademark/copyright symbols throughout the text, including ™ on page 22.

3. Your manuscript is missing the following sections: Introduction. Please ensure these are present, and in the correct order, and that any references to subheadings in your main text are correct. An outline of the required sections can be consulted in our submission guidelines here:

https://journals.plos.org/globalpublichealth/s/submission-guidelines#loc-parts-of-a-submission

4. We have noticed that you have uploaded Supporting Information files, but you have not included a list of legends. Please add a full list of legends for your Supporting Information files after the references list. 

5. Figs 1-4: please (a) provide a direct link to the base layer of the map (i.e., the country or region border shape) and ensure this is also included in the figure legend; and (b) provide a link to the terms of use / license information for the base layer image or shapefile. We cannot publish proprietary or copyrighted maps (e.g. Google Maps, Mapquest) and the terms of use for your map base layer must be compatible with our CC-BY 4.0 license. 

Additional Editor Comments (if provided):

Reviewers' comments:

Reviewer's Responses to Questions

**Comments to the Author**

1. Does this manuscript meet PLOS Global Public Health’s publication criteria? Is the manuscript technically sound, and do the data support the conclusions? The manuscript must describe methodologically and ethically rigorous research with conclusions that are appropriately drawn based on the data presented.

Reviewer #1: Yes

Reviewer #2: Partly

2. Has the statistical analysis been performed appropriately and rigorously?

Reviewer #1: Yes

Reviewer #2: No

3. Have the authors made all data underlying the findings in their manuscript fully available (please refer to the Data Availability Statement at the start of the manuscript PDF file)?

Reviewer #1: Yes

Reviewer #2: Yes

4. Is the manuscript presented in an intelligible fashion and written in standard English?

Reviewer #1: No

Reviewer #2: No

5. Review Comments to the Author

Reviewer #1: 1. result section:

Sociodemographic characteristics

“Four out of ten (38.1%) and nearly one-fifth (23.3%) of households were from the Oromia and SNNPR regions, respectively”.

This sentence is not clear it is difficult to catch all the paragraph. The remaining paragraph talking about Oromia and SNNPR only? Or about Ethiopia?

2. Discussion:

Paragraph 1 line number three says “open defecation being preferable due to unpleasant odors and heat from latrines”. What type of heat can be generated from latrine? Please explain it well.

3. Conclusion:

6th line of the paragraph, the word should is used. It looks like obligation. If it is kind of recommendation, please use another better word that makes it recommendation.

General:

1. The absence of page and line numbers makes difficult to indicate the exact location of major improvement areas in this manuscript.

2. Define SNNPR abbreviation upon first appearance in the text.

Reviewer #2: The paper is based on the spatial distribution pattern and associated factors with lack of access to sanitation services in Ethiopia, based on census data.

Due to the demonstrated impact of lack of basic sanitation access has on public health, especially in underdeveloped countries, this paper stands out in addressing the problem and identify areas where these deficits are increased and are major factors that leads to health inequities. Recognizing these areas would improve the cost-benefit access interventions, increasing the population reached by the interventions in countries where resources are extremely limited.

Despite this, the manuscript presents several issues that should be reviewed.

Three different spatial methods were used: Moran I, interpolation and scan statistics. But was not explained a clear conceptual framework that shows why the use of each of them and which particular outcome was expected.

Many information is just of local relevance, and with little or not relevance to other areas than Ethiopia. This is a global journal, so explain more in detail about socio-cultural features could help the reader to better visualize the problematic.

The manuscript is based in the EMDHS data, but some results and methodology is different. Please explain why those differences.

There is not a clear conceptual framework in the inclusion of the independent variables. Given the spatial dependency is the core of the manuscript, other spatial related variables could be included like environmental variables, road structure variables, distance to relevant points (i.e. cities).

If possible, the inclusion of more socio-cultural variables as etnias, level of education could reveal relevant associations.

Given spatial dependency was observed, to a proper analysis and to extract the fitted values of the predictive variables, the model should include in its structure the effect of the spatial dependency. Please revised the use of Conditional autoregressive (CAR) priors, or the Bayesian spatial modelling with INLA, both methods are suggested in this kind of analyses. Also, a spatial model with INLA would allow to perform a predictive model better than Krigging.

Even my native language is not English, I notice that throughout the text are observed sentences with none sense or redundant information, and also grammatical errors. Major revision of text should be performed to get a clear and easy going narrative.

Also, many references do not correspond to the cited data

Specific comments:

Lines:

16-20: In the abstract I suggest to avoid mentioning softwares and focus on mentioning the methodology. There are methods not included in the abstract.

24: avoid refer to figures in the abstract

25-27: The inclusion of the sense of the association, so the reader easily understand the findings

28: That households have limited sanitation service access it is not a conclusion of the current work, it is widely known and explained in the background section.

28-30: The conclusion is not in sense on the original research question. Also, focusing on the awareness of residents about this problem, places the responsibility on residents. For what I understood from the manuscript, the main outcome of the study is to identify areas with higher prevalence of lack of access to sanitation services and the associated factors. The conclusion should reflect the main conclusions from the obtained results, and the implications of it.

39-40: The sentence presents data as global information, but the references are from local data. Please rephrase the sentences so the reader can understand if is global or local data. If is global data, so the references should be improved

48: please explain if Jimma is an state, municipality, town, so de reader can understand

48: “45% of available latrine facilities are poor in hygiene, and the pipelines are poor in function [12]”. Where?

56: which feature of household head?

57: Toilet age?

57-58: rephrase to: “may represent associated factors to sanitation services access”

58-59: please rephrase, to maybe: "could be useful to localize specific areas with greater deprivation of sanitation access where prioritize sanitation interventions.

60: Please replace “highlight” to “describe”.

74-79: The sampling design is not completely understandable. So I went to read the EMDHS. I notice that many important details are missing here. So, the text should be extended and revised or derive directly to the reference.

76: how were distributed this 305 EA? How many EA are in the 21 stratas?

77: the sample selection was proportional to the population? Is not clear if the results showed are representative of the population.

83: what do you mean with zero coordinates?

84: what do you mean with had no proportional access to sanitation? are missing data or what?

87: The output variable is at household level? as yes or not?

93-94: how it was measured wealth status?

94: the category “Shared facilities” by its nature, it is not correlated with improved sanitation access? Also, in the EMDHS is one of the features that classifies the kind of sanitation access, which is the objective to include as a independent variable?

96: location of toilet facility, is not naturally related to the conditions of poor sanitation service?

118-124: the output of the variogram and the kriging should be included in result section or as supplementary material

122: The input data is at household or cluster level?

124: move the reference where corresponds

125-137: please specify test settings for statistical inference in Satscan. Also, please specify if in the output the overlapping was allowed and justify the decision.

130: The minimum and maximum spatial cluster size was based on what?

139: Is not being entirely clear if the model is at household or at cluster level. And, which distribution family? binomial? Negative binomial?

144: replace “variation” by “dependency” or correlation.

146: In the multilevel model, multicollinearity was tested?

147: which was the random factor? the cluster or the community?

155-163: The sociodemographic characteristics are from the sample population? It is representative of the studied population? What it means “under the rich wealth index” ? that 56.4% were not rich households? So almost 44% were?

164-168: please revise the paragraph, in the first sentences you reflect the proportion with improved sanitation, then the second without sanitation and in the third you relates percentages of both services referring to improved and unimproved services. Maybe the structure of the paragraph may be: first focus in the improved types of services, and then to the unimproved services.

171: limit precision to two decimals

173: convert to kilometers

175: This sentence should be in the figures subtitle or legend.

175: I interpret that the red color points indicate the area the area where households with poor sanitation services access are aggregated or clustered.

179-180: why open defecation is now treated as another category?

181: again, these results are representative of the study population? or are the sociodemographic features of the sample population? Age of whom? Sex of whom?

The percentage of household with sharing facilities appears inverted in the text.

183-193: In the way that the analysis was conducted it does not bring relevant information. I think that It should be avoided geographical overlapping in the output, stablish a greater minimum size and a smaller maximum size of cluster. Given that by the way that was conducted most of the territory is included in some cluster.

184: This sentence should be in the figure´s subtitle.

185-186: this is information is related to what? Are Id´s of clusters? It brings some relevant information?

187-190: is more useful to show this information only in the map as a figure.

Figure 1: this figure is likely a Moran I method explanation than a result, if relevant, should be in supplementary material. Improve subtitle to be self-explained

Figure 2: Improve the legend. Why open defections is a different class?. Improve region labels to be clearly read. Improve subtitle to be self-explained

Figure 3: Improve legend. what it means the value? Improve subtitle to be self-explained.

Figure 4: Improve legend.

About the figures in general, please adapt the scale to coincide between the three figures. Also, given the three figures are based on the same map of Ethiopia, they could be converted in one figure where appears the three maps.

Table 2: Please modify the orientation of the table.

218: “place of residency refers to what?

219: rephrase: "to appropriate sanitation service access"

224: “staying on-premises” means households that obtain their water on its own premise?

Table 3: improve the table tittle and explain the response variable

236: Some data appear different in the EDHS, for example the percentage of households with access to improved sanitation services in the rural area.

253-254: It is redundant to say that households in the significant clusters were more likely to be vulnerable to poor sanitation service access than households outside the window.

254-257: Please revise the text. The interpolation only showed the inputted information, didn´t confirmed.

261: Sharing toilet facilities is a classifier characteristic by the EMDHS to determine if the household has access to improved sanitation services or not, if you treated differently, you should justify and discuss it.

275: Please review the sentence, a connector is missing after “might be”.

282-284: The sentences is repeating information from the previous sentence.

286: what it means with “residency”.

301-302. Review the sentences, avoid “poor enough”. Maybe: “rural people may not Count with adequate financial resources and so are unable to access improved sanitation services”

305-311: Review the paragraph. Avoid repeating results in Conclusion section. Focus in bringing the outputs of the study, relevant information for the stakeholders, and relevant information for global public health academic community.

318-319. Rephrase the sentence. “specific vulnerable households spatially” does not have sense.

6. PLOS authors have the option to publish the peer review history of their article (what does this mean?). If published, this will include your full peer review and any attached files.

**Do you want your identity to be public for this peer review?** For information about this choice, including consent withdrawal, please see our Privacy Policy.

Reviewer #1: No

Reviewer #2: No

---

## [Editor Report · Decision Letter 1]

6 Mar 2023

Spatial and multilevel analysis of sanitation service access and related factors among households in Ethiopia: Using 2019 Ethiopian national dataset.

PGPH-D-22-01942R1

Dear Mrs. Addisalem Workie Demsash,

We are pleased to inform you that your manuscript 'Spatial and multilevel analysis of sanitation service access and related factors among households in Ethiopia: Using 2019 Ethiopian national dataset.' has been provisionally accepted for publication in PLOS Global Public Health.

Best regards,

Reginald Quansah, Ph.D.

Academic Editor